Downsizing a heavyweight: factors and methods that revise weight estimates of the giant fossil whale Perucetus colossus

http://orcid.org/0000-0001-5022-1053 Motani Ryosuke 1 rmotani@ucdavis.edu
http://orcid.org/0000-0003-4678-5782 Pyenson Nicholas D. 2 pyensonn@si.edu
1 Department of Earth and Planetary Sciences, University of California, Davis , Davis, California , United States
2 Department of Paleobiology, National Museum of Natural History, Smithsonian Institution , District of Columbia , United States
Farke Andrew
Electronic publication date: 2024 Feb 29
Publication date: 2024
Volume: 12
Electronic Location ID: e16978
Received 2023 Sep 21; Accepted 2024 Jan 29
Copyright: © 2024 Motani and Pyenson
Copyright year: 2024
Copyright holder: Motani and Pyenson
License: This is an open access article distributed under the terms of the Creative Commons Attribution License, which permits unrestricted use, distribution, reproduction and adaptation in any medium and for any purpose provided that it is properly attributed. For attribution, the original author(s), title, publication source (PeerJ) and either DOI or URL of the article must be cited.
License URL: https://creativecommons.org/licenses/by/4.0/

Keywords: Body size, Paleobiology, Estimation, Fossil record, Marine mammals

Funding: The authors received no funding for this work.

==============================
Extremes in organismal size have broad interest in ecology and evolution because organismal size dictates many traits of an organism’s biology. There is particular fascination with identifying upper size extremes in the largest vertebrates, given the challenges and difficulties of measuring extant and extinct candidates for the largest animal of all time, such as whales, terrestrial non-avian dinosaurs, and extinct marine reptiles. The discovery of Perucetus colossus, a giant basilosaurid whale from the Eocene of Peru, challenged many assumptions about organismal extremes based on reconstructions of its body weight that exceeded reported values for blue whales (Balaenoptera musculus). Here we present an examination of a series of factors and methodological approaches to assess reconstructing body weight in Perucetus, including: data sources from large extant cetaceans; fitting published body mass estimates to body outlines; testing the assumption of isometry between skeletal and body masses, even with extrapolation; examining the role of pachyostosis in body mass reconstructions; addressing method-dependent error rates; and comparing Perucetus with known physiological and ecological limits for living whales, and Eocene oceanic productivity. We conclude that Perucetus did not exceed the body mass of today’s blue whales. Depending on assumptions and methods, we estimate that Perucetus weighed 60–70 tons assuming a length 17 m. We calculated larger estimates potentially as much as 98–114 tons at 20 m in length, which is far less than the direct records of blue whale weights, or the 270 ton estimates that we calculated for body weights of the largest blue whales measured by length.

Introduction

Extremely large organisms, especially those belonging to vertebrates, readily capture public interest. However, these organisms are equally valuable to biologists because organismal extremes can be informative about fundamental ecological and evolutionary processes (Goldbogen, Pyenson & Madsen, 2023). Whales (or cetaceans) are particularly good examples of this dual phenomenon: living cetaceans, such as blue whales (Balaenoptera musculus) are perennial objects of popular fascination as the largest vertebrates ever; equally, studies on blue whale physiology, behavioral and feeding ecology have important implications for understanding broader trends among cetaceans and other mammals (Abrahms et al., 2019; Goldbogen et al., 2019a, 2019b). Recent improvements in animal-borne technology, including tags and aerial photogrammetry, have dramatically increased the precision for measuring the body sizes of living cetaceans in the field (Christiansen et al., 2019; Bierlich et al., 2022). These improvements in data collection facilitate further studies in the macroecology of these extremely large organisms: in many cases, body size informs energetics, and models of the ecological impact that these top consumers have on ocean ecosystem health (Savoca et al., 2021).

Despite the improved precision for measuring body size in living cetaceans, there remain challenges for validating and disseminating basic information about the body size of these taxa. In both popular and scientific literature, the specific body size values for superlative taxa are often cited as approximations or they are unspecific about their precision or origin. At closer inspection, the actual data for many of these superlative or champion size values are surprisingly difficult to find, vet, and evaluate.

Recently, Bianucci et al. (2023) reported a new basilosaurid whale, Perucetus colossus, based on a partial skeleton collected in Eocene rocks from southern Peru. Based on the associated skeletal material (i.e., thoracic and lumbar vertebrae in association with four ribs and an incomplete pelvis), Bianucci et al. (2023) suggested it was the heaviest whale ever, possibly reaching 17 to 20 m in length and 85 to 340 tons in total weight, rivaling or exceeding the mass of the largest blue whales. They based their estimates on a new method, in which they first estimated the total skeletal mass of Perucetus through extrapolation from the skeletal material and then used the value to secondarily extrapolate its body mass, assuming skeletal to body mass ratios based on extant cetaceans and sirenians. A simple ratio mandates an isometric relationship between body and skeletal masses, and they justified this step by testing for isometry using a phylogenetically controlled regression. They argued that the productivity in Eocene oceans was sufficiently high to support a marine predator weighing as much as 340 tons, a proposal with as profound implications for vertebrate morphology as for marine ecology.

The logic behind such a high body mass estimation may appear consistent within its own framework. However, their method involves questionable assumptions that suggest their body mass estimates are not reasonable when viewed from different perspectives. For example, a simple allometric relationship between whale length and mass suggests that the largest mass estimates for Perucetus cannot fit the body of a 20-m bauplan. A 17 to 20-m whale is much smaller than a 30-m whale (Figs. 1A and 1B vs. 1C and 1D), so for it to be as heavy as the latter, it would need to be 3.375 (=1.53) times denser or 1.83 times fatter in body diameter, or various combinations of increases in these two values. Yet, the whole-body density of vertebrates is known to fit in a narrow range of 0.75 to about 1.2 (Larramendi, Paul & Hsu, 2021), with the highest values found only in land turtles with heavy bone armor, whereas a whale can only be so much fatter than the blue whale. Therefore, it is extremely difficult for a 20 m whale to rival a 30 m individual in body mass.

Figure 1 Comparisons of body size between Perucetus colossus and Balaenoptera musculus based on Paleomass models with superelliptical exponent of 2.0.

(A) Perucetus with a fork length of 17 m, based on the lateral view only and thus overestimating the true volume. (B) Same with a fork length of 20 m. (C) B. musculus with a fork length of 30 m, based on both the lateral and ventral views, approximating the true volume. (D) B. musculus with a fork length of 30 m, based on the lateral view only and thus overestimating the true volume. (D) Overestimates the volume of (C). Scalebars are 30 m.

The body mass of fossil vertebrates is never directly measurable, so paleontologists have struggled to quantitatively estimate this value for over a century (Gregory, 1905). Today, there are many methods available, which are roughly group into regression-based and volumetric approaches (Hurlburt, 1999; Smith, 2002; Brassey, 2016; Campione & Evans, 2020; Motani, 2023). We note that all of these methods have challenges: for example, the best body mass estimates for terrestrial non-avian dinosaurs are derived from a cross-examination of the results from both classes of approaches (Sellers et al., 2012; Campione & Evans, 2020). The published mass estimates of Perucetus colossus have yet to be validated in this regard.

The purpose of this article is to cross-examine the published body mass estimates of Perucetus using data and methods informed by different perspectives. Specifically, we test the following hypotheses, which were assumed to be true by Bianucci et al. (2023): (1) the published body mass estimates of Perucetus fit the body outline as reconstructed in the same article; (2) the assumption of isometry between skeletal and body masses is valid even with extrapolation; (3) pachyostosis in Perucetus did not lead to overestimated body mass; (4) error rate is low in the body mass estimation method used; (5) skeletal to body mass ratio is well-established and stable within Cetacea; and (6) a large body mass of 340 tons was ecologically feasible given Eocene oceanic productivity.

Materials and Methods

Body silhouette images, numeric data, and R code used in the analyses described below are all available in Supplementary Information. All length and mass data were log10 transformed before regression.

Body mass of the largest blue whale

The body mass of the largest blue whale has never been measured, although there is a good basis for estimating this value. Total length (TL) has been historically used by cetologists and marine mammalogists since the early twentieth century, and it was widely adopted as a standard by the early 20th century whaling industry (Mackintosh & Wheeler, 1929), which had abundant access to large, intact cetacean carcasses. By contrast, cetacean body mass is extremely difficult to measure reliably for cetaceans heavier than ~100 kg (Gambell, 1970; Lockyer, 1976); total length is also more invariant to fluid loss and distortion once the specimen close to weighing equipment.

We used Lockyer’s (1976) compilation of records for TL and weights for large cetaceans as a starting point for the primary literature, along with select records published in a narrative by Small (1971:31–34). Following Scheffer’s (1974) line of reasoning for identifying the largest blue whale ever, we focused on females from the Southern Hemisphere (owing to sex and geographic size biases; Branch et al., 2007) and a timeframe before the mid-twentieth century conclusion of large-scale pelagic whaling. Accordingly, the longest reliably measured blue whale was a female 33.26 m long (109.12 feet long; see Risting, 1928). Winston (1950) enumerated the piecemeal weighing of a 27.1 m female blue whale that was at least 136.4 tons, although additional reputable, but less comprehensive records point to blue whales minimally weighing between 140 tons and 160.4 tons (Lockyer, 1976). While other methods exist to measure large cetacean body weights (see Gambell, 1970 for sperm whales [Physeter macrocephalus]), TL provides the most consistently comparable and widely sampled value for this aim.

For the purposes of this study, we used five body lengths—17 m that was used as the minimum length for Perucetus, 25 m that was used by Bianucci et al. (2023), 30 m as convenient benchmark for blue whale length maxima, 33 m after Risting (1928), and lastly an arbitrary value of 15 m (half the latter benchmark) for data exploration purposes—to validate the implication by Bianucci et al. (2023) that Perucetus was larger than the largest blue whales. We employed both length-based on volumetric approaches so that the results can be cross-examined. For length-based estimation, we used the data set Nishiwaki (1950) and Ohno & Fujino (1952), who collectively reported the body mass and length of 34 individuals of blue whales, spanning from 21.3 to 27.1 m in length. The individual mentioned by Winston (1950), whale No. 319, is included in Nishiwaki’s (1950) dataset. Of the 34, we removed three pregnant individuals and retained the rest. Lockyer (1976) added some data to those mentioned above but we did not include these data because the weight data summarized in that work were inconsistently collected and the sample size is small. We ran an ordinary least square analysis on the data using R (R-Core-Team, 2020), with body mass as dependent and length as independent variables. We used the resulting regression relationships to extrapolate the body mass of a blue whale with a body length of 15, 17, 25, 30, and 33 m, with confidence and prediction intervals (Table 1).

Table 1 Body mass estimates of blue whales (Balaenoptera musculus) from regression-based and volumetric approaches.

Fork	Regression-based mass estimates (ton)	Volumetric mass	
length	Assuming 7% blood loss	Assuming 14% blood loss	Estimates (ton)	
(m)	Mean	95% CI	95% PI	Mean	95% CI	95% PI	Mean	Min	Max	
15	19.0	[13.6–26.6]	12.7–28.4	20.5	[14.7–28.7]	13.7–30.7	25.6	25.0	26.2	
17	28.6	[22.4–36.7]	20.5–39.9	31.0	[24.2–39.7]	22.2–43.1	37.3	36.4	38.2	
25	101	[97–106]	81–127	110	[104–115]	87.4–137	119	116	121	
30	184	[157–217]	140–242	199	[170–234]	151–262	205	200	210	
33	252	[201–316]	183–346	272	[217–342]	198–374	273	266	279	
Note:

CI and PI refer to confidence and prediction intervals, respectively.

Data from Nishiwaki (1950) and Ohno & Fujino (1952) were collected by cutting large whale carcasses into measurable pieces (as described by Winston, 1950), leading to blood (and other fluid) loss, despite the best efforts to retain them for weighing. Therefore, we sought to account for blood loss in our regressions. There is a general relationship between body to blood volume ratios and deep diving in whales (Bernaldo De Quirós et al., 2019). Blue whales are not deep diving taxa (Schreer & Kovacs, 1997), and non-deep-diving cetaceans fit in a range of 7 to 14% for this ratio. We therefore used this range of ratios to bracket for blood loss.

The logic of estimation explained above is not fundamentally different from one used by Lockyer (1976), who proposed two equations for estimating blue whale body mass from TL. However, we did not use these equations because one has an unspecified degree of blood loss compensation embedded, whereas the other is based on mixed data from inconsistent methods. The equations also lack confidence and prediction intervals.

For the volumetric approach, we used an R package Paleomass (Motani, 2023), which brackets the true body mass of marine vertebrates using 3D models built from orthogonal silhouettes. The method has a high accuracy, with a mean absolute error of 1.33% when the silhouettes are accurate. We built Paleomass models based on a dorsal view photograph (https://commons.wikimedia.org/wiki/File:Blue_Whale_001_body_bw.jpg) and lateral view reconstruction (https://commons.wikimedia.org/wiki/File:Balaenoptera_musculus_NOAA.jpg) derived from NOAA (National Oceanic and Atmospheric Administration), and assumed a whole body density of 1.027 g/cm3, which is the density of seawater at surface (Stewart, 2008). We used a superelliptical exponent range of 2.0 to 2.3, which was found to be suitable for cetaceans (Motani, 2023). Given that the two images are from different individuals which are most likely smaller than 30 m, the error level is probably higher than 1.33%. Nevertheless, it provides sufficient accuracy to cross-examine the length-based estimates, which tends to have a much larger error margin.

Volumetric mass estimation of Perucetus

It is impossible to accurately estimate the body mass of Perucetus volumetrically, given the lack of accurate body outline reconstructions from both ventral and lateral views. However, it is still possible to bracket the upper bound of its body mass based on the data from Bianucci et al. (2023). There are two difficulties involved: that only the lateral view was reconstructed (Bianucci et al., 2023; Fig. 2), with the vertebral column curled down toward the tail, and that the accuracy of the published body outline reconstruction is unknown.

Figure 2 Straightening of the vertebral column from published reconstruction of Perucetus colossus.

(A) Modified reconstruction through straightening of the body axis along the vertebral column, in black lines, used in this study. (B) Trace of the original reconstruction of Perucetus from Fig. 2 of Bianucci et al. (2023), in blue lines, showing a curved vertebral column. Lines for the body axis were traced along the centers of vertebral centra. See Materials and Methods.

We addressed the first difficulty in two steps. First, we straightened the published body reconstruction of Perucetus colossus along the vertebral column, as seen in Fig. 2. Lines perpendicular to the vertebral column were drawn at the center of every second vertebral centrum, spanning the dorsal and ventral margins of the body (Fig. 2). Then the vertebral column was straightened posteriorly by rotating each segment from the previous step, leaving some curvature near the fluke that approaches similar curves in extant whales (Buchholtz, 2001). The ends of the perpendicular lines were connected to draw the new dorsal and ventral margins (Fig. 2, black curves). Second, we used the straightened lateral view outline to bracket the upper bound of the body in the R package Paleomass. Extant cetaceans have a more slender profile in dorsal view rather than lateral view (Bierlich et al., 2022), so using the lateral view in place of ventral view constantly leads to overestimation of the true body volume (compare Figs. 1C and 1D). The largest distortion is caused by the tail stock, which is strongly compressed laterally. We tested this observation with Paleomass based on the blue whale silhouettes discussed above, as well those from three other species in Motani (2023). The overestimation rate was 26% in harbor porpoises (Phocoena phocoena), 23% in Heaviside’s dolphins (Cephalorhynchus heavisidii), 14% in bottlenose dolphins (Tursiops truncatus), and 5% in blue whales. In particular, blue whales have an exceptionally low value because they have heads sufficiently large and flattened to counterbalance the tail, which is laterally compressed, so using the value of 5% for non-mysticetes would likely lead to errors. We therefore assume that the body mass of Perucetus was overestimated by 14 to 26% by using only the lateral body silhouette.

The second difficulty cannot be addressed easily. However, it is evident that the published body outline reconstruction of Perucetus is already very fat for its length—its body depth, assuming a body length of 20 m, is already as deep as that of a 30 m blue whale (Figs. 1B vs. 1C). We therefore infer that the published reconstruction is close to the upper limit of how fat Perucetus could have been for its length. In combination with the factors explained above, the resulting volumetric estimate of body mass is expected to overestimate true mass, allowing bracketing of the upper bound.

Perucetus was estimated to be 17 to 20 m in total length, a value that is well-constrained by comparison with closely related basilosaurid whales (e.g., Basilosaurus) that are known from complete skeletons e.g. (Gingerich, Smith & Simons, 1990). We therefore used this range for volumetric estimates of body mass. We also experimented with slightly smaller sizes of down to 15 m, which is an arbitrary decision, but nonetheless useful for the purposes of exploring possible outcomes for estimating body mass.

We assumed a density of surface seawater (1.027 g/cm3) for Perucetus, as with blue whales. Although marine tetrapods alter their body density during the course of a day, depending on their behavior, the body density range always includes neutral buoyancy (Domning & de Buffrénil, 1991; Watanabe et al., 2006; Aoki et al., 2011; Sato et al., 2013; Gordine, Fedak & Boehme, 2015). This density consistency is partly because marine tetrapods need to breathe at sea surface, during which they are neutrally or positively buoyant. We note that this consistency is also true for pachyostotic marine mammals, such as sirenians (Domning & de Buffrénil, 1991; Kipps et al., 2002; Lefebvre, 2023). Therefore, our use of the surface seawater density is justified.

Effect of assumed isometry in extrapolation

Bianucci et al. (2023) tested the isometry between skeletal and bone masses with a positive result and used this result as the justification for using a simple ratio instead of an allometric regression line to estimate body mass. However, the test of isometry only demonstrates that the difference between the isometric line through the mean, representing a constant ratio, and the corresponding allometric regression line is very small in the range of observation. Notably, it does not guarantee that the two are identical, especially outside of the data range. It is therefore necessary to examine if the isometric line from a ratio and the corresponding allometric regression lines are sufficiently close to each other outside of the data range.

We took the following steps in examining this question. We started by downloading the data and R scripts used by the authors from their GitHub site (https://GitHub.com/eliamson/ColossalCode). The site did not contain the phylogenetic tree used in computation, named MamPhy_fullPosterior_BDvr_DNAonly_4098sp_topoFree_NDexp_MCC_v2_target.tre, so we located it in another GitHub site (https://github.com/n8upham/MamPhy_v1/tree/master/_DATA) and downloaded it. Upon examination of the tree, however, we noticed that the topology of Mysticeti placed Megaptera at the base of the clade, a relationship that is not supported by current phylogenetic consensus (e.g., McGowen et al., 2020). We therefore used Timetree.org to download a time-calibrated phylogeny of Cetacea in Newick format to be used as an alternative tree. The tree is provided in Supplemental Information.

We made four subsets of data, representing terrestrial mammals, Cetacea, Odontoceti, and Mysticeti, respectively, and regressed the body mass against skeletal mass for each subset, with Phylogenetic Generalized Least Square (PGLS). This step is the opposite of Fig. 4 in Bianucci et al. (2023) because the aim is to estimate body mass from skeletal mass. We used the same tree as Bianucci et al. (2023) for terrestrial mammals, and the above stated trees for cetaceans. We then plotted isometric lines, i.e., of slope 1, though the means of these four subsets, respectively. Lastly, we assessed the differences between isometric and allometric lines in the part of the graph extrapolated to accommodate Perucetus.

Effect of pachyostosis

The data used by Bianucci et al. (2023) contained 11 individuals belonging to Florida manatees (Trichechus manatus) selected from Domning & de Buffrénil (1991), which clearly have much lighter body mass for their skeletal mass compared to cetaceans. Manatees are known for pachyostosis, which thickens cortical bones that gives them greater density than in typical vertebrates (Houssaye, 2013). It is reasonable for pachyostotic mammals to have lighter body mass for skeletal mass because they are not necessarily much fatter than non-osteosclerotic species despite the added bone weight. Given that Perucetus was also pachyostotic, it is possible that it also had a lighter body mass for skeletal mass compared to modern cetaceans. We therefore estimated the body mass of Perucetus based on the regression relationships of skeletal mass against body mass in manatees. We used ordinary least square regression given that the data are monospecific.

The original data of Domning & de Buffrénil (1991) contained 64 individuals from two species of manatees (Trichechus spp.), of which 50 had both body and skeletal mass data. Twelve of the 50 were less than 1.5 m in body length and considered perinates according to Ingle & Porter (2020), leaving 38 individuals in the complete manatee data. Bianucci et al. (2023) selected the largest 11 individuals of the 38, with body lengths greater than 3 m. The intention may have been to include only adults but this threshold value is too large—Domning & de Buffrénil (1991) used this value for a threshold for “large adults” (p. 337), citing Odell (1981), who said sexual maturity was reached at a body length of about 2.6 m in females and 2.75 m in males. More importantly, such subsetting of data based on size range often adds bias in the results of allometric regression, especailly if it is used for extrapolation (see Fig. S1 for how adult-based regression lines depart from the general trend when extrapolated to a whale size, potentially causing errors). We therefore tried including all 38 individuals in regression and compare the results with those from truncated data set with only 11 individuals.

Error rates of body mass estimation methods

Bianucci et al. (2023) did not test the accuracy of the new body mass estimation methods based on skeletal to body mass ratio. We estimated the body mass of cetaceans in their data with these methods and calculated the mean absolute error for each, together with the minimum and maximum absolute errors. We also use the methods to predict the body mass of the only pachyostotic marine mammal with a large sample size (manatees), and calculated the same error statistics. For comparison, we repeated the same with regression-based estimation methods.

Skeletal to body mass ratios in Cetacea

Bianucci et al. (2023) used the skeletal to body mass ratios calculated for six individuals of large whales from five species in a previous study (Robineau & de Buffrénil, 1993). However, these ratios are based on estimated body masses, casting some doubts on the accuracy. They also added Delphinus delphis from de Buffrénil, Collet & Pascal (1985) and seven species of smaller cetaceans. We therefore cross-examined the skeletal to body mass ratios using the wet bone mass data from other authors (Nishiwaki, 1950; Omura, 1950; Ohno & Fujino, 1952; Fujino, 1955; Omura et al., 1969).

Wet bone masses are substantially larger than dry bone masses that were used to calculate the skeletal to body mass ratio (de Buffrénil, Collet & Pascal, 1985), yet they are expected share scaling trends if the latter is approximated as a proportion of the former. Then, comparing the scaling patterns of the two may allow identification of outliers.

We fitted linear models to the data from the literature mentioned above to test if different species share a common intraspecific trend when regressing wet and dry bone masses against body mass. We tested the correlation between bone mass and body mass with a class variable, where species are separated into categories and, if a single species had both wet and dry bone mass data, they were also treated as separate categories. We also separated Antarctic and North Pacific populations of sperm whales into different categories within this class variable, given that they have vastly different bone masses for body mass. Lastly, we calculated the interspecific trend based on species means with PGLS to examine if it differed from the intraspecific trend.

Consensus between regression-based and volumetric mass estimates

For each mean body mass estimate from PGLS for Perucetus, we tested if it was compatible with any volumetric estimates from within the body length range of 15–20 m.

Results

Body mass of the largest blue whale

Using a linear least square regression on log-transformed body length and mass, we calculated that a 33 m blue whale would weigh 234 (95% CI [187–294]) tons. Assuming that body fluid equivalent to 7% of the body volume was lost, the body mass of the whale would be 252 (CI [201–306]) tons and the value is 272 (CI [217–342]) tons with a ratio of 14%. In combination, the mass is estimated to be in the range of 252–272 (CI [201–342]) tons with the regression approach. The volumetric approach resulted in a body mass range of 266–279 tons, overlapping the regression-based confidence intervals. Although these volumetric estimates lack accuracy because of the factors discussed in Materials and Methods, it is sufficiently close to regression-based results to allow validation of the claim that the largest blue whales are in the vicinity of 270 tons. See Table 1 for corresponding prediction intervals and estimates at other body lengths.

Volumetric mass estimation of Perucetus

Based on the lateral view only, the body mass of a 20 m Perucetus is estimated to be about 98 to 114 tons. Results, including estimates for other body lengths, are summarized in Table 2.

Table 2 Volumetric estimates of body mass for Perucetus colossus with different body lengths.

Fork length (m)	Volume from lateral view
(m3)	Corrected volume
(m3)	Body mass (ton)	
20.0	120–126	95.4–111	98.0–114	
19.0	103–108	81.8–94.9	84.0–97.5	
18.0	87.7–92.0	69.6–80.7	71.5–82.9	
17.0	73.8–77.5	58.6–68.0	60.2–69.8	
16.0	61.6–64.6	48.9–56.7	50.2–58.2	
15.0	50.7–53.3	40.3–46.7	41.3–48.0	

Effect of assumed isometry in extrapolation

The results of PGLS extrapolation and its comparison with isometric extrapolation are given in Fig. 3 and Table 3. Although the differences among isometric lines and allometric regression lines may appear minor when viewing the entire span of the data (Figs. 3A and 3B), the actual bias in the extrapolated range is obvious when magnifying the part that is relevant to Perucetus (Fig. 3C). In this range, a minor deviation between lines results in a large difference in body mass because of the nature of log scale. For example, an isometric line results in body masses that are 100 to 150 tons larger than those from cetacean PGLS line when the minimum cetacean ratio is assumed, and 17 to 26 tons with the mean cetacean ratio (Fig. 3; Table 3). These two ratios were used by the original authors, so the published body masses were inflated by the assumption of isometry.

Figure 3 Regression lines for body mass against skeletal mass based on different subsets of the data analyzed, illuminating how minor differences in regression coefficients or assumption leads to substantial body mass differences of more than 100 tons.

(A) Phylogenetic generalized least square regression of body mass against skeletal mass based on the data from Bianucci et al. (2023). (B) Ordinary least square regression of body mass against skeletal mass based on manatee data. (C) Same as A and B combined but zooming into the black rectangle, which is the part relevant to Perucetus colossus. Red horizontal rectangle extending from the y axis represents the volumetric estimates of body mass at a fork length of 20 m (above), 18 m (middle), and 15 m (below), based on the body outline reconstruction of Bianucci et al. (2023). Blue vertical band from the x axis denotes the range of skeletal mass estimated for Perucetus by Bianucci et al. (2023), with a solid line inside being the mean estimate. Double-headed arrows on the right are body mass ranges suggested by: a, based on skeletal mass assuming isometry, as in Bianucci et al. (2023); b, based on regression from 11 individuals of manatee used by Bianucci et al. (2023); c, based on regression from 38 individuals of manatee in the original data of Domning & de Buffrénil (1991); d, the body outline reconstruction by Bianucci et al. (2023) at a fork length of 20 m; e, the same at a fork length of 18 m; and f, the same at a fork length of 15 m. See Tables 1 and 2 for the values. S/B refers to Skeletal mass/Body mass ratio. White arrow indicates the direction toward the unknown regression line for pachyostotic whales.

Table 3 Regression-based estimates of body mass for Perucetus colossus from estimated bone mass.

	With Min Est. Bone Mass	With Mean Est. Bone Mass	With Max Est. Bone Mass	
	Mean	CI	PI	Mean	CI	PI	Mean	CI	PI	
Cetacea min S/B ratio	238	—	—	291	—	—	343*	—	—	
Cetacea mean S/B ratio	152	—	—	185*	—	—	219	—	—	
Cetacea max S/B ratio	105	—	—	128	—	—	150	—	—	
Cetacea PGLS	135	—	—	163	—	—	193	—	—	
Manatee OLS truncated	61.4	[2.74–134]	2.70–142	74.2	[2.91–189]	2.84–194	86.6	[3.06–245]	2.99–251	
Manatee OLS complete	40.0	[20.9–76.8]	172–93.0	47.4	[24.2–93.1]	20.1–112	54.6	[27.3–109]	22.7–131	
Note:

S/B refers to Skeletal mass/Body mass ratio. Asterisk indicates values used by Bianucci et al. (2023) to set the body mass range of Perucetus.

Effect of pachyostosis

The effect from pachyostosis on body mass estimation is not negligible. When using the regression line from manatees, the body mass of Perucetus is estimated to be 45.3 (35.3–52.1) tons based on a data set with 50 individuals (Table 3). These values are more than 100 tons less than 163 tons from the cetacean PGLS line, which does not consider the effect from pachyostosis.

Truncation of the dataset also led to a large difference in body mass estimates: for example, compare 74.2 (61.4–86.6) tons based on 11 individuals with the above-mentioned results. When comparing the regression results from the complete versus truncated manatee data sets, truncated data set gave less stable results than the complete one. The confidence and prediction intervals from the truncated data set are too wide whereas those from the complete data set are much narrower (Table 3).

Error rates of body mass estimation methods

The mean absolute error from each method is summarized in Table 4, together with the maximum and minimum absolute errors. The ratio-based methods, assuming isometry between the skeletal and body masses, generally have high error rates even when applied to the data from which the ratios were derived. The worst performance is when using the minimum skeletal to body mass ratio from cetaceans, which led to a mean absolute error of 61% and maximum absolute error of 127.9% even when applied to modern cetaceans from which the ratio was derived. The maximum error was 287% when applied to pachyostotic marine mammals. This method was used by Bianucci et al. (2023) to arrive at the maximum body mass of 343 tons. Regression-based methods generally performed better, although the error rates are not low.

Table 4 Absolute errors from various estimation methods when applied to the data on which each is based, as well as to pachyostotic marine mammals.

		With own data	With pachyostotic data	
	n	Mean	Min	Max	Mean	Min	Max	
Cetacea min S/B ratio	14	61.2%	0.0%	127.9%	174.6%	2.3%	287.0%	
Cetacea mean S/B ratio	14	17.9%	0.2%	45.3%	80.0%	11.4%	146.7%	
Cetacea max S/B ratio	14	29.3%	0.0%	56.1%	27.7%	0.9%	69.8%	
Cetacea PGLS	14	18.9%	5.9%	35.3%	79.9%	13.2%	150.3%	
Terrestrial PGLS	40	14.1%	0.6%	38.7%	18.3%	0.0%	66.1%	
Manatee OLS truncated	11	13.6%	5.4%	31.1%	18.6%	1.7%	58.9%	
Manatee OLS complete	38	18.9%	0.5%	81.2%	18.9%	0.5%	81.2%	
Note:

S/B refers to Skeletal mass/Body mass ratio.

Skeletal to body mass ratios in Cetacea

A linear model found a common intraspecific trend of 0.858, which was strongly significant (p-value < 2 × 10−16). Interspecific trends from PGLS were 1.02 (p < 2 × 10−16) for dry bone mass, and 1.11 (p < 2 × 10−16) for wet bone mass. Therefore, the intra- and interspecific trends are opposite, i.e., larger whales tend to have less skeletal mass for body mass within a single species whereas larger whale species tend to have more skeletal mass for body mass compared to smaller species. The interspecific trend is shared with patterns known for terrestrial mammals (Prange, Anderson & Rahn, 1979).

Distribution of species on the bone versus body mass space is given in Fig. 4. Data for blue whales exhibited a strange behavior. This species has the largest wet bone mass for body mass given the trend among whales examined, yet it has the smallest dry bone mass for body mass, again given the trend. It is unknown if this reflects the reality or is an artifact of the estimations involved in the dry bone data that were pointed out in Materials and Methods.

Figure 4 Skeletal mass plotted against body mass for cetaceans.

Wet skeletal masses have symbols infilled with colors, whereas dry skeletal masses have empty symbols without a fill color. Colored convex hulls were drawn for species with sample sizes larger than two.

Consensus between regression-based and volumetric mass estimates

As seen in Fig. 3C and Tables 1 and 2, mass estimates from cetacean regression lines are too large to be validated by volumetric estimates, even at the maximum possible body length of 20 m (red horizontal band with arrow d in Fig. 3C). Therefore, it is impossible to pack the mass suggested by these lines into a whale that is 20 m or shorter. Within the body length range of 17–20 m suggested by Bianucci et al. (2023), only those regression-based estimates from the truncated manatees and terrestrial mammals yield a consensus result with volumetric estimates. Notably, estimates based on truncated manatee data matched the volumetric estimates assuming a body length of 18.0 m (overlap of arrow b and e in Fig. 3C). When considering body size smaller than 17 m, there was a match between volumetric estimates from a 15-m individual and regression-based estimates from the complete manatee data set (overlap of arrows c and f in Fig. 3C).

Discussion

Bianucci et al. (2023) proposed that the middle Eocene basilosaurid Perucetus ranks among the largest vertebrates to have ever lived. Based on estimation methods from incomplete skeletal material, they calculated body masses for Perucetus as large or larger than extant blue whales. We argue that there are clear methodological factors that likely inflated their published body mass estimates for Perucetus (Table 5). The factor leading to greatest potential overestimation was the assumption of isometry between the body and skeletal mass, which resulted in upwards of about 100 to 150 tons of overestimation (Fig. 3C; Table 3). This factor is followed by the neglect of the impact of pachyostosis on body versus skeletal mass relationships, which likely added another 100 tons of overestimation, although the exact amount is open to discussion. The effect of this latter factor is not only logical but also empirically evident from the fact that Perucetus body size reconstructions based on non-pachyostotic whales are too large to physically fit into a bauplan with 20 m or less in total length (Fig. 3C; Tables 1 and 2). Perucetus would never be as heavy as the largest blue whale even if its body was packed with only compact bones, i.e., making its body density twice as much as the appropriate value. Moreover, the published upper bound of 340 tons can only be reached by assuming that a pachyostotic whale has a skeletal to body mass ratio of a whale with the skinniest skeleton for body mass, which directly contradicts the premise for the rest of the study by Bianucci et al. (2023). Therefore, it is likely that the published body mass range of 180–340 tons (mean and maximum estimations) is vastly overestimated.

Table 5 Major causes of error and their effects, which may not be mutually exclusive. An asterisk (*) indicates that effects may overlap among these factors.

Cause of bias	Resulting overestimation	Ref.	
Unreasonable assumption of isometry between body and skeletal masses during extrapolation	Up to 100 to 150 tons	Fig. 3B	
Use of modern whale regression when pachyostotic animals have more bones for body mass*	About 100 tons	Fig. 3B	
Inclusion of whales with unusually light dry bone mass for body mass*	About 10 tons	Fig. 3B	
Unexplained exclusion of data points from manatee data	10 to 20 tons	Fig. 3B	
Low accuracy of the methods	Varies	Table 4	
Use of mysticete tree that is not current	Small	Text	
Body to skeletal mass ratio for large whales calculated using estimated rather than measured body mass*	Not quantifiable	Text	
Assumption that Perucetus is accurately modeled based on Cynthiacetus	Not quantifiable	Text	

Body mass estimation from skeletal to body mass ratio is sensitive to a minor deviation in the ratio, especially when applied to extremely large vertebrates. This sensitivity, coupled with a high degree of intraspecific variation in this ratio (see Domning & de Buffrénil, 1991) would result in a large overestimation. This ratio, for example, varies between 2.7 and 5.5% in 40 individuals of common dolphin Delphinus delphis examined by de Buffrénil, Collet & Pascal (1985) based on dry bone mass and the scatter is greater in adults (see Fig. S1). When examining a broader interspecific dataset of several cetacean species, the ratio varies between 2.2 to 5.1% (see Bianucci et al., 2023), and this range results in more than 200 tons of body mass difference for Perucetus (Table 3). Similarly high or even greater variations are known in the ratios based on wet mass (Fig. 4). For example, sperm whales (Physeter macrocephalus) show a substantial difference in this ratio based on wet bone mass between Antarctic and North Pacific populations (Fig. 4), with ratios 12.2–15.5% in Antarctic and 8.3–11.1% in North Pacific. Thus, estimating body mass of one population based on the skeletal to body mass ratio in the other population would result in errors of up to 100 tons at the size range of Perucetus. While we note that the body size of Perucetus was not estimated based solely on wet skeletal mass in cetaceans, these observations cast a serious doubt against the accuracy of body mass estimation based on this ratio when applied to species of extinct giant cetaceans, especially based on incomplete material.

There are other factors behind the method that led to the published body mass range of 180–340 tons (Table 5). First, the accuracy of skeletal mass estimates for large extant cetaceans is unknown, given that both the skeletal and body mass were roughly estimated in the data of Robineau & de Buffrénil (1993) used by Bianucci et al. (2023)—bone masses were measured by compensations for metal support and the holes left by it is based on estimation that was not explained. Second, the phylogenetic tree used was not current, although this bias had a small effect. Lastly, the bone volume of Perucetus was based on scaling and dilation from a much smaller species, Cynthiacetus peruvianus, whereas the accuracy of this procedure is unwarranted. This assumes a close similarity of basic anatomy between the two taxa apart from size, which is not well established, given the unique features of Perucetus. Also, this process is the first of the double extrapolations: based on this extrapolation, another extrapolation was made to estimate the mass, lowering the accuracy.

We maintain that the data at hand are too limited to estimate the body mass of Perucetus accurately, and the limited accuracy of available estimation methods (Table 4) adds to the problem. At the same time, some estimates are more likely than others. Paleobiologists estimating body mass have demonstrated the value of achieving a consensus among different methods, including at least one from regression-based estimation in addition to a volumetric approach. For Perucetus, the only mass estimates that satisfy such a consensus of methods yield a range of 71.5–82.9 tons, which resulted between volumetric estimates assuming a body length of 18 m and regression-based estimates from the truncated manatee data set (Tables 1 and 2; Fig. 3C, arrows b and e). However, the regression line for a truncated data set has questionable validity. Notably, this latter approach yields estimated body masses that are higher than those of extant cetaceans with a body length of 20 m—the heaviest blue whales and fin whales at this body length would be less than 60 tons if we applied the same methods as with the largest blue whales (Nishiwaki, 1950). When we used a regression line for manatees from a complete data set (i.e., untruncated), the resultant body mass estimates match the range from volumetric estimation assuming a body length of 15 m (our arbitrary lowest total length value, half the benchmark of 30 m), in the range 41.3–48.0 tons (Tables 1 and 2; Fig. 3C, arrows c and f). This finding suggests that there is some validity for contemplating a size class below the lower limit of body length range of 17 m, which Bianucci et al. (2023) produced for Perucetus based on scaling from Cynthiacetus. The 17 m size estimate, however, carried many assumptions, including that the vertebral counts are identical between Cynthiacetus and Perucetus. We argue that the validity of these assumptions are still open to debate, pending the discovery of more material belonging to Perucetus. In that view, we think that the lower range of 41.3–48 tons is probably not unreasonable, but note that these are still heavier than the masses of modern whales at the same length—a 15-m sperm whale is less than 40 tons (Omura, 1950; Ohno & Fujino, 1952). Also, a 17 m Perucetus would be about 41.3–48 tons if it is less fat than reconstructed by Bianucci et al. (2023) by about 17% in diameter. It is worth mentioning that the body sizes of basilosaurids, such as Perucetus, have been the subject of detailed size estimations because many of their skeletons are very completely known (e.g., Basilosaurus and Cynthiacetus). Gingerich (2016) estimated that a 15 m long B. isis weighed 5,840 kg (also see Marino et al., 2000; Marino, McShea & Uhen, 2004; Uhen, 2004), approximately the weight of today’s killer whales (Orcinus orca), although this estimate may have a wide range of imprecision (see Gingerich, 2016). At about 9 m in length, Cynthiacetus had a smaller body mass than Perucetus, but it was among the larger basilosaurids, which have a range in body sizes (Corrie & Fordyce, 2022; Antar et al., 2023).

Finally, there are two additional factors that cast doubt on the extremely large body mass estimates provided by Bianucci et al. (2023) for Perucetus. The first factor is based on feeding energetics grounded in principles of comparative physiology, which suggests that a 340 ton cetacean could not maintain homeostasis nor sustain itself metabolically. Goldbogen et al. (2019b) demonstrated that biomechanical and ecological factors limit body size extremes in extant cetaceans. For the largest filter feeding cetaceans, blue whales are limited by the prey escape from their oral cavity: at sizes beyond 33 m, hypothetically supersize blue whales would be unable to close their mouths quickly enough to overcome the escape response of their prey, and then trap and filter them (Potvin, Goldbogen & Shadwick, 2012). There is no evidence to suggest that Perucetus was a filter feeder; until cranial material is discovered, the best inference is that Perucetus consumed single prey items (e.g., fish or processed parts off items larger than its oral cavity) as other basilosaurids did (Uhen, 2004). If so, it is likely that Perucetus conformed to the energetic allometries that constrain extant odontocetes, which feed on single prey items (Goldbogen et al., 2019b). Sperm whales (Physeter macrocephalus) represent the largest size maximum for single prey feeding cetaceans at about 20 m and about 80 tons in weight, although the bauplan of Physeter differs in significantly from that of basilosaurids (see above).

Today, blue whales migrate to areas where their prey seasonally aggregate in high enough densities that enable a high return on the energetic investment (Goldbogen, Pyenson & Madsen, 2023); there are no known prey densities of krill nor any other zooplankton that are high enough to sustain mammalian energetics beyond what we observe today. Single prey feeding odontocetes use the bathypelagic ecosystem for hunting, and their consumption rates of deep-sea fish and cephalopods are constrained by breath holding for these feeding dives. Overall, it is not clear that a 340 ton cetacean could dive, breathe or maintain a heart rate based on the known allometries for the largest cetaceans today (Goldbogen et al., 2019a).

The second external factor relates to food webs and ocean productivity. Other basilosaurids were carnivorous and fed on fish and other basilosaurids (Uhen, 2004; Fahlke, 2012; Voss et al., 2019). While there are no cranial remains for Perucetus, there is no reason for suspect it was not carnivorous, despite the pachyostosis in the available postcranial material that resembles the same condition in most fossil and extant sirenians (de Buffrénil et al., 2010). The largest herbivorous marine mammal was Hydrodamalis, a recently extinct dugongid sirenian that reached 9 m in length and weighed 8–10 tons (Domning, 1978); its large body size compared to other sirenians was likely an adaptation to cold water kelp forest ecosystems (Estes, Burdin & Doak, 2016). Both marine angiosperms and marine macroalgae have low preservation potential, but their presence can be inferred from the sirenian fossil record (Vélez-Juarbe, 2014). Unfortunately, there are no reported Paleogene age sirenians nor seagrasses in South America. Thus, we see the preponderance of evidence weighing strongly against the hypothesis that Perucetus was the first known herbivorous cetacean.

Bianucci et al. (2023) presented three hypotheses for the feeding ecology of Perucetus: i) it was herbivorous; ii) it was a benthic feeder; and iii) it was a scavenger. The first hypothesis lacks evidence (as detailed above), and invokes an unlikely possibility: Perucetus as the only cetacean lineage to evolve herbivory. Such a scenario has happened in carnivorans (see Figueirido et al., 2010), and repeatedly in terrestrial tetrapods (Sues & Reisz, 1998), which makes it unlikely but a possible argument, pending the discovery of cranial and dental remains. The evidence for the second hypothesis derives from the pachyostosis of the postcranium of Perucetus, which is analogous to sirenians (see above as well). As Bianucci et al. (2023) point out, gray whales (Eschrichtius robustus) are capable of benthic feeding, as are humpback whales (Megaptera novaeangliae; see Ware et al., 2014), but both taxa lack recognizable osteological specializations for this feeding style, and retain the fundamental feeding apparatus for filter-feeding in the water column instead of the benthos (see Pyenson & Lindberg, 2011). Overall, this hypothesis is likely the strongest of the three. The last hypothesis, benthic scavenging, is hard to test given the current data for Perucetus, but it is theoretically possible: large body size would benefit a marine scavenger on the tradeoff of foraging vs. consuming, while optimizing the efficiencies of metabolic rate and transport costs, even for a mammal (Ruxton & Houston, 2004). The one challenge: obligate marine scavengers, large or small, endothermic or ectothermic, have not been identified in today’s ecosystems.

Regardless of its feeding mode, Perucetus was likely top consumer in Eocene marine food webs. It is unclear if Eocene coastal marine ecosystems of Peru were as productive as these nearshore regions have been in the late Neogene. Middle Eocene oceans retained most of the features from the Early Eocene Climatic Optimum (Norris et al., 2013), with limited ocean mixing and nutrient delivery at lower latitudes, and marine prey communities dominated by pelagic fish and sharks. In this vein, Perucetus was collected in rocks with pelagic teleosts, which are present along with other presumably pelagic stem cetaceans from the Paracas Formation (Uhen et al., 2011), and the relatively warm water paleoenvironmental reconstruction by Malinverno et al. (2021) points to a weak upwelling system. The comparatively longer food chains of Early Eocene oceans, in contrast to those of the Neogene, led to energy loss between trophic levels, which in turn limited the overall size and diversity of top predator populations (Norris et al., 2013). In this regard, Perucetus was likely a top consumer on body size alone, but it remains unclear what its role may have been in Middle Eocene nearshore ocean food webs.

Conclusions

Bianucci et al. (2023) argued that the largest size estimates of Perucetus challenged many assumptions about organismal extremes, including a range of body weight that matched or exceeded the weight of today’s blue whales. We examined a series of factors and methodological approaches that included: body size data sources from large extant cetaceans; fitting published body mass estimates of Perucetus to its proposed body outlines; testing the assumption of isometry between skeletal and body masses for cetaceans; examining the role of pachyostosis in biasing body mass reconstructions; addressing method-dependent error rates; and comparing Perucetus with physiological and ecological limits determined from extensive studies of living whales, along with Eocene oceanic productivity. Overall, we conclude that Perucetus did not exceed the body mass of today’s blue whales, and more closely matched the body size and weight of sperm whales.

We presented a series of size estimates, dependent on specific assumptions and methods to estimate that Perucetus weighed at a minimum in the range of 41.3–48 tons, and at a maximum 98–114 tons. We think the assumptions required for estimating body weights of >100 tons for Perucetus are unrealistic. Rather, we suspect the likeliest body weights would point to an approximately 60–70 ton size range, which is already about 10–11 times larger than weight estimates for its close relative, Basilosaurus. Ultimately, the discovery of more fossils belonging to the skeleton of Perucetus, especially cranial and dental material, will play an important role in testing these estimates, as well as hypotheses about the ecological role of the largest cetacean yet known in Eocene oceans.

Supplemental Information

Supplemental Information 1 Body mass regressions, body mass data, body outline images, and phylogenetic trees.

We thank Geerat Vermeij for reading an earlier version of this manuscript.

Additional Information and Declarations

Competing Interests

Author Contributions

Data Availability

Nicholas D. Pyenson is an Academic Editor for PeerJ.

Ryosuke Motani conceived and designed the experiments, performed the experiments, analyzed the data, prepared figures and/or tables, authored or reviewed drafts of the article, and approved the final draft.

Nicholas D. Pyenson analyzed the data, authored or reviewed drafts of the article, and approved the final draft.

The following information was supplied regarding data availability:

The body mass and skeletal weight and code are available in the Supplemental Files.

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
