# Peer review of "Downsizing a heavyweight: factors and methods that revise weight estimates of the giant fossil whale Perucetus colossus"

_PeerJ, doi:10.7717/peerj.16978_

## Round 0.1 · original submission · Minor Revisions

Thank you for your patience in awaiting the reviews--all of the reviewers returned their comments in a timely fashion, but it took some extra effort for me to find multiple reviewers. Two of the three reviews suggest relatively minor modifications to the paper. The remaining review is from authors of the original paper on Perucetus, invited per PeerJ policy because your manuscript critiques areas of their paper. They have a number of requests for clarification, correction, or adjustment, which should be addressed in detail in your revised manuscript and/or response document. In particular, note the methodological points and clarifications--although some may potentially be differences of opinion, others should be clarified or modified in the updated manuscript. I recognize that methods of body mass estimation are a topic of endless nuance and sometimes debate, so appreciate your efforts for revision.

·

Basic reporting

The English is fine, with very few typos, though some ambiguous formulations do not correctly refer to previous work (see comments below).
The literature list is ok. In some cases, the authors inadequately question the validity of results from previous works (see detailed comments below).
The article is well organized. However, questions about the validity of results from several parts will probably lead to a reorganization of the ms. The same is true for some figures, that may have to be deleted in a revised version of this work. See detailed comments below.
The submission is ‘self-contained. But see below for the validity of part of the results.

Experimental design

The primary research is within aims and scope of the journal.
The submission clearly defines the research question, which is be relevant and meaningful.
We found a relatively high number of flaws in this ms, and we question part of the methodologies used, as well as misleading and/or unfair formulations when criticizing previous works (see detailed comments below).
In several instances, we had to look in the raw data to understand some of the decisions taken by the authors (see comments below).

Validity of the findings

The support to part of the conclusions is weak (see comments below).

Additional comments

This review has been done jointly by Eli Amson and Olivier Lambert, two co-authors of the original work on Perucetus colossus. This paper focuses on the body mass estimates obtained by Bianucci et al. (2023) for this giant basilosaurid, highlighting possible limitations and mistakes in these estimates, while it does not discuss any of the methods used to obtain skeletal mass estimates for Perucetus, another central part of the work. Though we were at first sight very interested and curious about potential improvements of our own calculations, we quickly spotted a relatively high number of methodological issues, as well as inadequate formulations that do not accurately reflect the original meaning of many parts of the Bianucci et al. paper. Some of the most important methodological issues and inaccuracies are listed here below, followed by a long list of other moderate to more minor problems. While a few isolated parts of this manuscript may be worth publishing, we think that the main results are too heavily biased. Correcting these sections would probably result in a considerably different work, with significantly modified conclusions.

Major issues:

1- Volumetric mass estimation of Perucetus:
The authors surprisingly propose to estimate the mass of this extinct animal with a method generally used for extant taxa and/or fossil specimens with a preserved body outline, but based here on the lateral outline of an artist's reconstruction. As this reconstruction was not informed by any volume consideration, the whole approach starts from a very wrong premise. One may possibly say that the body shape in the reconstruction could be too thin compared to the proposed range of mass estimates, but doing the opposite to obtain mass estimates is certainly wrong.

2- Effect of pachyostosis:
- l. 252: 'Bianucci et al. (2023) selected the largest 11 individual Florida manatees of the 50, with body lengths greater than 3 m. this selection was never explained as far as we are aware.':
This is actually stated twice in the paper: 1- Main text, Methods: "mean of adults from ref. 26"; 2- Supplementary Table 22's footnote reads "mean of adults, based on Domning & Buffrénil 1991". 3 m is the minimum length of adults as defined in Domning & Buffrénil 1991).
- l. 254: 'We therefore tried including all 50 individuals in regression and compare the results with those from truncated data set with only 11 individuals.':
Why are you assuming that ontogenetic allometry of skeletal mass in Trichechus helps making such an estimation in Perucetus?
- l. 321: 'Truncation of the dataset also led to a large difference in body mass estimates': This is not surprising, as the authors compare a static allometry regression ("truncated" dataset) that uses only adults (as defined in Domning and Buffrénil (1991), the initial publication) to an ontogenetic allometry regression. We do not see the meaning of comparing these two sets of results here.
- l. 417: 'from a complete data set (i.e., untruncated)': Again, the full dataset includes neonate individuals. Neonates (especially of marine mammals) are known to have very different body tissue proportions than adults.

3- Skeletal to body mass ratios in Cetacea:
l. 340: ‘Therefore, the intra- and interspecific trends are opposite, i.e., larger whales tend to have less skeletal mass […]’: Again, it is unclear to us why confronting ontogenetic (or an attempt at doing so, see below) to evolutionary allometries is relevant here. Even if it was the case, the exact sample composition for each species, on which this "intraspecific trend" is based, is not specified in the text. One has to open the code and raw data to find out that the groups are very unequal, with some species represented by one individual and others with various ontogenetic compositions represented. It is therefore a mixture of static and ontogenetic allometries, as well as single observations, that is referred to here as "intraspecific trend". Doing so, we found in the code that for the interspecific pgls, the mean of adults was taken for Delphinus delphis (defined as Age >= 12 for males and >= 7 for females). Is that explained somewhere? For the pgls only the adults of Delphinus delphis are included. Why is then the entirety of the sampling for the other species (some of which also comprise several ontogenetic stages) considered appropriate?
Furthermore, opening the raw data (either 'Buffrenil_EA_1985.csv', or 'Whale_mass_length.csv.csv', for the first Delphinus specimen that we found (Specimen 1041 from Buffrénil et al. 1985), there was a copy/paste typo. The rest of the data should be reviewed carefully.

4- Error rates of body mass estimates and effect of isometry assumption:
- l. 335: 'This method was used by Bianucci et al. (2023) to arrive at the maximum body mass of 343 tons.': Of course, there is a huge error when one applies the minimum cetacean ratio to a manatee. Referring to this as "This method was used by Bianucci..." is misleading.
- l. 336: 'generally performed better': This is not surprising, as using a regression (and not “assuming isometry”, as it is put here) will dampen differences (inter- or intraspecific). Whether that is "better" to estimate the body mass of an extinct animal strikingly differing from extant cetaceans is not straightforward.
We would like to stress here the fact that because the studied animal is so different from any other vertebrate at the level of its skeleton, our goal was to remain as cautious as possible for our body mass estimates. This resulted in a relatively broad interval, in which the highest values should not be considered individually (as it is done in the present paper, see below). - Finally, just a comment: Bianucci et al. base their estimates on a single individual, which is very different from the selection of the largest individual among many for an extant species. There is a good chance that the holotype of Perucetus colossus is not the largest ever individual of its species, and this should probably be taken into account when formulating comparisons with the blue whale.


Other moderate to minor issues:

- l. 71: 'assuming skeletal to body mass ratios based from cetaceans': This is also based on sirenians.

- l. 84: 'Yet, the whole-body density of vertebrates is known to fit in a narrow range...': This may seem obvious. As mentioned in Bianucci et al., no living vertebrate approaches the degree of BMI seen in Perucetus.

- l. 97-...: 'we test the following hypotheses, which were assumed to be true ...': This formulation is misleading, as none of these statements were explicitly assumed in Bianucci et al. For example, Bianucci et al. mention a high seafloor productivity during the Bartonian, but they do not write that this was sufficient to support an animal weighing 340 tonnes, as they propose a broad range of body weight estimates (between 85 and 340 tonnes) without indicating any preference for the highest limit of this interval.

- l. 99: 'The published body mass of Perucetus fits...': This does not make sense, as Bianucci et al. did not provide a single mass estimate, but rather a broad range.

- l. 133: 'validate the statement by Bianucci et al. (2023) that Perucetus was larger than the largest blue whales': This has not been stated in this paper.

- l. 144: 'extrapolate the body mass of a blue whale with a body length of 17 ... m': 25 m is listed above. Which one is correct?

- l. 150: 'whether they are deep divers or not (Bernaldo De Quirós et al., 2019)': This paper states "there is only one measurement in a deep-diving species:the sperm whale". That is ca 20%. But this is actually an estimate (see ref cited therein). Also, there is another deep diving cetacean included therein, the short-finned pilot whale (Globicephala macrorhynchus), for which 12%, was estimated. The simple shallow/deep diver difference presented here does not seem as straightforward.

- l. 196: 'We therefore assume that the body mass of Perucetus was overestimated by 14 to 26% by using only the lateral body silhouette': We do not understand how it is possible to assume here the same bias as in cetaceans with no BMI. More broadly, this is the core issue here: a cetacean with the skeletal morphology of Perucetus must have had a very different overall body morphology from that of extant cetaceans.

- l. 202. 'We therefore infer that the published reconstruction is close to the upper limit of how fat Perucetus could have been for its length.' This sounds arbitrary. We do not understand what this inference is based on.

- l. 209: 'We also experimented with slightly smaller sizes of down to 15 m.': This also sounds fully arbitrary. What is the basis for this body length?

- l. 268: 'and seven species of smaller cetaceans whose derivation was not explained as far as we are aware.': This is actually explained, in the main text, methods: "The extant cetacean data were taken from Buffrénil et al.71, Buffrénil72..."

- l. 273: 'they are expected share scaling trends': Was this previously demonstrated?

- l. 286: 'For each mean body mass estimate from PGLS for Perucetus, we tested if it was compatible with any volumetric estimates from within the body length range of 15-20 m.': Is that assuming a sea water density? If so, why? Was this measured in any animal with BMI?

- l. 300: 'See Table 1 for corresponding prediction intervals and estimates at other body lengths.': Why are the pgls outputs not provided?

- l. 313: 'so the published body masses were inflated by the assumption of isometry.': This is misleading: no isometric equation is needed to use extant taxa's skeletal fraction to estimate BM. Furthermore, we did not find any details describing the PGLS outputs (except for later in the text, one finds a slope (referred to as a trend) of 1.02 for dry bone mass, and a mysterious “p=0”), and no test of whether the obtained slope deviates from isometry.

- l. 338: 'linear model found a common intraspecific trend of 0.858': Why is this called a trend and not a slope?

- l. 369: 'The factor contributing the largest degree of overestimation': How is that an overestimation? One should have access to the actual weight of the animal to make such a clear-cut claim.

- l. 370: 'which resulted in about 100 to 150 tons of overestimation': This (and other references in text and Table 3) is misleading, as it only refers to one of the cetacean skeletal fractions, the minimum one. "The assumption of isometry", when compared to the mean or maximum cetacean skeletal fraction, differs from 17-43 tonnes from the results here obtained with pgls.

- l. 378: 'coupled with a high degree of intraspecific variation in this ratio (see Domning and de Buffrénil, 1991)': This is a bit vague. Do you mean all specimens, including juveniles that can be as light as 20 kg?

- l. 380: 'in 40 individuals of common dolphin Delphinus delphis examined by de Buffrénil, Collet & Pascal (1985)': Again, when mixing all age classes, from early juveniles to adults, no surprise to see a large intraspecific variation.

- l. 382: 'The ratio varies between 2.2 to 5.1% in the cetacean samples of Bianucci et al. (2023)': Those are several species, so it is misleading to state this just after the previous sentences that refers to intraspecific variations.

- l. 387: 'Thus, estimating body mass of one population based on the skeletal to body mass ratio in the other population would result in errors of up to 100 tons at the size range of Perucetus.': This is misleading, as the mass of Perucetus was not estimated based on wet skeletal mass. Wet skeletal mass has been identified as metric prone to strong bias as early as 30 years ago (Robineau and Buffrénil 1993).

- l. 393: 'published body mass range of 180-340 tons': This is not the correct interval.

- l. 394: 'given that both the skeletal and body mass were roughly estimated in the data of
Robineau & Buffrénil (1993)': That is unfair. Of course, body masses of large cetaceans are estimated, like everywhere (including here). But the skeletons were directly weighted.
- l. 396: 'bone masses were measured by compensations for metal support and the holes left by it is based on estimation that was not explained': Another unfair statement. Why are holes referred to here? The metal supports referred to in Robineau & Buffrénil (1993) concern the limbs and skull. This looks like an unjustified criticism of this previous publication.

- l. 398: 'bone volume of Perucetus was based on scaling from a much smaller species': This is not accurate, as it obscures part of the procedure employed therein to perform this estimation (which did not only involve scaling).

- l. 405: 'the limited accuracy of available estimation methods': What would be the accuracy threshold the authors deem appropriate to permit BM estimations?

- l. 418: 'assuming a body length of 15 m, in the range 41.3-48.0 tons': This body length seems fully arbitrarily chosen, as no explanation is provided. This is especially worrying considering that the resulting estimates are among those chosen to be included in the abstract.

- l. 420: 'including that the vertebral counts are identical between Cynthiacetus and Perucetus': This is absolutely wrong. See Supplementary Methods of Bianucci et al. (2023), p. 11-13.

- l. 429: 'Gingerich (1998) estimated that B. isis weighted 5840 kg': Note that later Gingerich (2016) specifically refers to Basilosaurus BM estimate as questionable.

- l. 437: 'blue whales would be unable to close their mouths quickly enough to overcome the escape response of their prey, and then trap and filter them': Not sure to see how this is relevant to basilosaurids.

- l. 447: 'despite the pachyostosis in the available postcranial material that resembles the same condition in trichechid sirenians': And other families of sirenians as well.

- l. 451: 'Because there are no marine angiosperm nor macroalgae fossils in the Paracas Formation': This unfair statement fully overlooks the fact that these organisms rarely fossilize (especially in these conditions). Obviously, the absence of plant/algae fossils does not mean those were not present in the area (see for example Vélez-Juarbe, 2014). Furthermore, the authors only discuss here one of the three hypotheses proposed by Bianucci et al. for the feeding strategy and prey types of this animal.
Vélez-Juarbe, J. (2014). Ghost of seagrasses past: using sirenians as a proxy for historical distribution of seagrasses. Palaeogeography, Palaeoclimatology, Palaeoecology, 400, 41-49.

- l. 456: 'The late Eocene oceans retained most of the features from the Early Eocene': Perucetus originates from Bartonian levels, which means middle Eocene, not late.

- l. 459: 'Perucetus was collected in rocks with pelagic teleosts': See comments by Bianucci et al. about potential transport of the whale's carcass before deposition and burial. They find it more likely that this animal was living in a more coastal environment, as proposed for several other basilosaurids, and in comparison with extant sirenians.

- l. 470: 'Bianucci et al. (2023) argued that the largest size estimates of Perucetus challenged many assumptions about organismal extremes': Which assumptions?

Reviewer 2 ·

Basic reporting

This paper addresses the work Bianucci et al 2023 , who named a new species of basilosaurid (Perucetus colossus), who claimed that this remarkable animal potentially weighs more than the blue whale, the largest animal ever known. As Carl Sagan said, "extraordinary claims require extraordinary evidence", and the authors of this current study believe that the evidence put forward in Bianucci et al 2023 is insufficent and show in detail how it is unlikely to have been the case.

The paper is well written using clear, concise language. The introduction gives ample context regarding the benefits of accurately estimating the size of extremely large animals, beyond the intrinsic appeal of knowing what is the biggest. It is logically structured, going through the separate approaches used by the authors to dissect the claims of Bianucci et al 2023, and clearly stes the hypotheses tested. The figures are adequate and the R scripts provided in the ESM are easy to follow.

Experimental design

The research is well within the scoope of the journal, and as mentioned above the research questions are well defined, with hypotheses clearly stated. The methods are detailed and will permit study replication with relative ease.

Validity of the findings

As best as I can tell, the statistical anlyses employed here are sound, e.g. controlling for phylogeny where necessary. All underlying data has been provided where possible, and the data that cannot be provided due to copyright issues have information on how to them. The conclusions are supported by the results and sum up the paper accurately and succinctly.

Additional comments

This is a well written paper that shows in detail how it is unlikely that Perucetus colossus was as heavy as first proposed by Bianucci et al 2023. I believe it will be a welcome addition to the literature, and would like to see how this pans out in future, either with a response to this paper or by the finding of more complete material that would permit more refined estimates. I recommend the paper be accepted pending the fixing of a few small issues noted in the attached PDF.

Annotated reviews are not available for download in order to protect the identity of reviewers who chose to remain anonymous.

·

Basic reporting

I found this to be a well written manuscript about the potential pitfalls of the recent publication describing Perucetus. I found the introduction to be a very good primer to the field of body size estimation and to the more specific problem at hand. The figures were, overall, well produced, and are relevant to the content of the manuscript. I have no major concerns with the basic reporting in the manuscript.

Minor table comment:
1) Table 1: "CI", "PI", and "t" should be defined in the caption ("t" could also be replaced with "ton" as in the caption for Table 2).

Minor figure comments (ordered by figure #):
1) Figure 1: I believe "C overestimates the volume of B" should be "C overestimates the volume of B. musculus"?
2) Figure 2: I would suggest specifying (albeit how obvious it is) what the blue and black colors correspond to.
3) Figure 3: "d, the body outline reconstruction by the same" unclear what "by the same" means here.
4) Figure 3: The white arrow is not described in the caption.
5) Figure 4: The colored convex hulls (?) are not defined in the caption.
6) Figure 5: I would recommend removing the underscores in the legends (and non taxonomic information could/should be in parentheses).

Experimental design

I found the methodology to be very thorough and rigorous. The authors tested/addressed a number of potentially problematic areas related to the estimation of the body mass of both modern blue whales and the Eocene cetacean Perucetus. I found the Methods section to be well written, clear, and well organized.

Validity of the findings

I have no major comments regarding the validity of the findings.

Minor data comment:
1) It appears that the masses (and maybe other dimensions) are not consistent in units across the different supplemental files. For example, body mass appears to be log transformed in "Whale_mass_length.csv.csv", whereas it appears to be not transformed in "Domning_Buffrenil_1991.csv". I understand that the size data are from other primary sources, but it would be good to ensure unit/transformation consistency across these files and/or provide metadata with the supplement.

Additional comments

It was a pleasure reviewing this manuscript. I appreciate your thoroughness in addressing this, and I would be more than happy to read a revised version of the manuscript if needed.

---

## Round 0.2 · accepted · Accept

Thank you for your thorough response to all of the reviewers' comments; in reading your responses, I believe that all concerns have been sufficiently incorporated or rebutted. Thus, I believe that this manuscript is ready for publication.

In proof, I would suggest consideration of two very minor details, which do not rise to the level of requesting another round of minor revisions. Line numbers refer to the revised PDF. These points are:

- line 360/361: Along with the reviewers, I also find the "p=0" to be puzzling; yes, it is very closely approximated to 0, but it's highly unlikely to be truly zero (even if that's what the software spits out). I personally prefer to use p<10*-16 (or some other small value), because it more correctly and accurately reflects the likely situation.
- line 502: italicize "Perucetus"